# Characterisation of Non-Pathogenic Premutation-Range Myotonic Dystrophy Type 2 Alleles

**DOI:** 10.3390/jcm10173934

**Published:** 2021-08-31

**Authors:** Jan Radvanszky, Michaela Hyblova, Eva Radvanska, Peter Spalek, Alica Valachova, Gabriela Magyarova, Csaba Bognar, Emil Polak, Tomas Szemes, Ludevit Kadasi

**Affiliations:** 1Institute of Clinical and Translational Research, Biomedical Research Center, Slovak Academy of Sciences, 845 05 Bratislava, Slovakia; kadasi@fns.uniba.sk; 2Department of Molecular Biology, Faculty of Natural Sciences, Comenius University, 841 04 Bratislava, Slovakia; michaela.hyblova@gmail.com (M.H.); bogeecs@gmail.com (C.B.); polakemo@gmail.com (E.P.); tomas.szemes@uniba.sk (T.S.); 3Comenius University Science Park, 841 04 Bratislava, Slovakia; 4Geneton s.r.o., 841 04 Bratislava, Slovakia; 5Department of Neonatology, Faculty Hospital Nove Zamky, 940 34 Nove Zamky, Slovakia; eva.radvanszky@gmail.com; 6Centre for Neuromuscular Diseases, University Hospital Bratislava, 826 06 Bratislava, Slovakia; peter.spalek@seznam.cz; 7Department of Clinical Genetics, Faculty Hospital Trencin, 911 71 Trencin, Slovakia; alica.valachova@fntn.sk; 8Department of Laboratory Medicine, Faculty Hospital L Pasteur Kosice, 041 90 Kosice, Slovakia; gabriela.magyarova@unlp.sk; 9Grow Cube s.r.o., 900 84 Igram, Slovakia

**Keywords:** cellular retroviral nucleic acid-binding protein, *CNBP*, DM2, expansion, myotonic dystrophy type 2, premutation

## Abstract

Myotonic dystrophy type 2 (DM2) is caused by expansion of a (CCTG)_n_ repeat in the cellular retroviral nucleic acid-binding protein (CNBP) gene. The sequence of the repeat is most commonly interrupted and is stably inherited in the general population. Although expanded alleles, premutation range and, in rare cases, also non-disease associated alleles containing uninterrupted CCTG tracts have been described, the threshold between these categories is poorly characterised. Here, we describe four families with members reporting neuromuscular complaints, in whom we identified altogether nine ambiguous *CNBP* alleles containing uninterrupted CCTG repeats in the range between 32 and 42 repeats. While these grey-zone alleles are most likely not pathogenic themselves, since other pathogenic mutations were identified and particular family structures did not support their pathogenic role, they were found to be unstable during intergenerational transmission. On the other hand, there was no observable general microsatellite instability in the genome of the carriers of these alleles. Our results further refine the division of *CNBP* CCTG repeat alleles into two major groups, i.e., interrupted and uninterrupted alleles. Both interrupted and uninterrupted alleles with up to approximately 30 CCTG repeats were shown to be generally stable during intergenerational transmission, while intergenerational as well as somatic instability seems to gradually increase in uninterrupted alleles with tract length growing above this threshold.

## 1. Introduction

Short tandem repeats (STRs), or microsatellites, are responsible for a great part of interindividual and intraindividual genomic variability, and they belong to the most polymorphic regions of the human genome [1]. Their physiological and pathophysiological significance is of immense importance, since they are connected not only to the fine regulation of expression levels of many genes [2], or they regulate their alternative splicing [3], but they also have severe pathological conditions, mainly associated with dynamic expansions of STR loci causing serious neurological and neuromuscular diseases. In repeat expansion disorders, expandable repeat tracts may be located in functionally diverse parts of different genes [4]. In these, moreover, not only the onset of symptoms, but also their specific constellation as well as their severity tend to depend on the repeat numbers and/or on the presence of interruptions in the repeat motifs themselves. One of these disorders, myotonic dystrophy type 2 (DM2; OMIM_#602668), is connected to expansion of a (CCTG)_n_ repeat tract located in the first intron of the *cellular retroviral nucleic acid-binding protein* (*CNBP*; also called *zinc finger protein 9, ZNF9*) gene. Although the expansion has been known since 2001 [5], the normal variation of the DM2-associated STR locus in the *CNBP* gene is still poorly characterised, requiring further annotation [6]. The expandable (CCTG)_n_ repeat is part of a complex repetitive motif (TG)_n_(TCTG)_n_(CCTG)_n_, while in non-disease-associated alleles, the (CCTG)_n_ part of the repeat tract is generally interrupted by one or more GCTG, TCTG or ACTG tetraplets. Alleles having uninterrupted CCTG parts were, however, also described [5,6,7,8]. The largest explicitly published non-disease-associated allele contained 26 CCTG repeats with two interruptions being part of a 176 bp complex tract [5]. The most commonly cited repeat number for the lower boundary of pathogenicity is 75 [5], although two patients with 55 and 61 CCTG repeats were also reported with no equivocal proof of the pathogenicity of these alleles [7]. However, to date, no studies have characterised the normal, premutation- and mutation-range *CNBP* alleles in detail. Since both DM2 full expansions as well as premutations were found to be relatively common in the general population [6,9], a more detailed establishment of which allele sizes and structures predict phenotypic expression of DM2 and which are benign alleles seems to be important for clinical counselling as well as to understand the origin and population dynamics of the DM2 expansions [10]. In this paper, therefore, we report and annotate four uninterrupted CCTG repeat-containing *CNBP* alleles belonging to the poorly described grey zone, which most likely do not cause disease in themselves but are inter-generationally unstable and can thus be considered for DM2 premutations.

## 2. Patients and Methods

Our findings are based on results of routine molecular genetic testing performed during a differential diagnostic process of patients clinically suspected to have myotonic dystrophy, but in whom the molecular testing resulted in an alternative diagnosis. The pathogenic/benign nature, as well as the stability of the identified grey-zone CCTG alleles, was characterised in four families, including several members in at least two generations of each family. Besides these relatively rare alleles, the characterisation of the stability of more common CCTG alleles was performed by the evaluation of the molecular genetic testing results of a larger number of patients in whom testing for DM was performed. 

### 2.1. Patients and Families

All four families, which are at the centre of our study, were referred for DNA diagnostics because of certain neuromuscular complaints in at least one member of each family (marked as index cases). Altogether, blood samples were analysed from 22 individuals of these families (Figure 1). These included eight members spanning four generations in the first family, four members spanning three generations in the second family, three members in three generations in the third one and three members in two generations in the fourth family. Genomic DNA was isolated from leukocytes by a PuregeneTM DNA Purification Kit (Qiagen, Düsseldorf, Germany) or by phenol–chloroform extraction for some older samples. Informed consent consistent with the Helsinki Declaration was obtained from each subject before DNA testing. 

### 2.2. Myotonic Dystrophy Molecular Testing

The DM1 and the DM2 repeat loci were characterised using the singleplex as well as the multiplexed versions of conventional PCR and repeat-primed PCR according to the previously described protocols [11,12,13]. Briefly, PCR products were separated by capillary electrophoresis on the ABI Genetic Analyzer 3130xl (Applied Biosystems/Life Technologies, Foster City, CA, USA), while sizing of the fragments was ensured using the high-density GeneScan 1200 LIZ dye Size Standard (Applied Biosystems/Life Technologies). Raw data were further analysed using the GeneMapper Software version 3.2 (Applied Biosystems/Life Technologies). Complex tract length was calculated from conventional PCR results while the number of uninterrupted CCTG repeats from repeat-primed PCR results was found, counting the highest peak at the end of the resulting peak distribution as the estimator of the repeat number [12].

### 2.3. Genotyping of the rs1871922 Polymorphism

Genotyping of the single nucleotide polymorphism rs1871922 was performed by Sanger sequencing as previously described [6]. Briefly, following preamplification using the published primer pairs (ZNF9_seq-for 5′-TCT GAT TGG ACT GCC GAA C-3′ and ZNF9_seq-rev 5′-CCA GGA CGA AGA AAG GAC AG-3′) and purification of the PCR products using Exonuclease I/Shrimp Alkaline Phosphatase treatment, we performed sequencing applying the BigDye Terminator v1.1 cycle sequencing kit (Applied Biosystems/Life Technologies) and the ABI Genetic Analyzer 3130xl (Applied Biosystems/Life Technologies). Raw data were further analysed using the Sequencing Analysis v5.4 software (Applied Biosystems/Life Technologies). Visualisation of the sequencing reads, assembly and variant calling was performed using the ChromasPro v1.6 software (Technelysium Pty Ltd., Queensland, Australia).

### 2.4. Assessing the Stability of Common Interrupted CNBP Alleles

Stability of interrupted *CNBP* alleles during intergenerational transmission was assessed by re-evaluation of our results from routine molecular diagnostic testing of 290 cases/families. Out of these, 27 families were suitable for the evaluation of possible complex tract length changes, resulting in altogether 57 recordable parent-to-child transmission events while one parent/one child was counted for one transmission, two parents/one child for two transmissions, one parent/two children for two transmissions, etc.

### 2.5. Assessing the Stability of Other Microsatellite Loci

Stability of other microsatellite loci in the genome of the carriers of the described unstable *CNBP* alleles was assessed through the evaluation of the DM1 CTG repeat in the *DMPK* gene, as well as 16 other microsatellite loci included in the Investigator ESSplex SE Plus Kit (15 STR markers in the new European Standard Set (ESS) of loci, plus SE33) (QIAGEN, Hilden, Germany). The *DMPK* alleles were analysed as described above. Characterisation of the ESS/SE33 loci was performed according to the manufacturer’s instructions with a subsequent evaluation of PCR products by capillary electrophoresis on the ABI Genetic Analyzer 3130xl (Applied Biosystems/Life Technologies, USA) calibrated with BT5 matrix. Conditions for capillary electrophoresis were optimised according to the manufacturer’s recommendations. Raw data were further analysed using GeneMapper Software version 3.2 (Applied Biosystems/Life Technologies, USA).

## 3. Results

To assess the stability of common alleles of the complex *CNBP* repeat tract during intergenerational transmission, we retrospectively analysed *CNBP* repeat tract lengths of interrupted alleles in 57 recordable parent-to-child transmission events. We found no length changes of the repeat tracts bearing interrupted CCTG alleles (Appendix A). On the other hand, during our diagnostic process, we identified uninterrupted grey-zone CCTG alleles in the *CNBP* gene, which represented possible premutation-range alleles. These were initially identified in one affected member from family_1, in the only affected member of family_2, in the only affected member of family_3 and in the asymptomatic sister of the only affected member of family_4 (Figure 1, Table 1). Since these alleles fell into the range of poorly characterised grey-zone alleles while they were identified in families with neuromuscular complaints, it was necessary to exclude their pathogenic/causative nature in each family for correct counselling purposes. Extramuscular symptoms were not systematically examined in all reported individuals and family members, but if those complications were manifested in the patients and recorded in their health records they were included among the symptomatology.

The index patient from family_1, a 31-year-old woman suffering typical myotonic dystrophy symptoms, underwent DNA testing ordered by a clinical geneticist because of her positive DM family history. Her first reported symptoms, upper extremity weakness and hand grip myotonia, appeared when she was approximately 18 years old. Later, she developed weakness in both proximal and distal leg muscles as well as in cranial muscles together with myopathic face with down-slanting mouth corners. This patient refused electromyographic investigation. DNA analyses revealed a clearly pathogenic-range CTG expansion in the *DMPK* gene and an uninterrupted 33 CCTG-containing *CNBP* allele (190 bp complex tract). She inherited the expanded CTG allele from her maternal grandfather and the 33 CCTG allele from her maternal grandmother. The maternal grandmother carried a 32 CCTG allele but was reported to have no DM-related complaints at the time of sampling, i.e., when she was 80 years old. These findings indicated one-unit extension of the uninterrupted tract during transmission through two generations. The mother of the proband died at 27 from pulmonary embolism, before the family came to our attention. The only son of the proband inherited neither the DM1 expansion nor the DM2 premutation.

A 34 CCTG allele (202 bp complex tract) was identified in a 43-year-old man, the only affected member of family_2, who was also noted in our previous report [6]. He developed his first symptoms approximately at the age of 10. Current clinical examination revealed adequately configured musculature, while the patient suffers from hand grip myotonia, disabling myotonic leg muscles stiffness causing gait problems with the typical “warm-up” phenomenon. Percussion myotonia was noted in the region of the thenar and needle electromyographic testing of resting interosseous and rectus femoris muscles revealed characteristic myotonic discharges. The uninterrupted allele was maternally inherited and also transmitted to the subject’s son, while the mother of the index case had 33 and the son 35 CCTG repeats. The proband’s DNA was originally isolated from blood samples and deposited when he was 21 and repeatedly when he was 26 and 43 years old. Analysis revealed the same allele length in all isolations, thus suggesting time-dependent somatic stability at least in his leukocytes. Besides the uninterrupted *CNBP* CCTG tract, a well-known nonsense variant (NM_000083.2:c.2680C > T; p.Arg894*) in the chloride voltage-gated channel 1 (*CLCN1*) gene was identified in his genome in a homozygous state. Observations that neither of the parents was reported to suffer from neuromuscular symptoms, while both of them were heterozygous carriers of the p.Arg894* *CLCN1* variant and the mother also of the premutation-range CCTG allele provide molecular confirmation of recessive *myotonia congenita*, which is in line with clinical, molecular and genealogical findings in the patient/family.

Although the clinical signs of the only affected member of family_3, a 47-year-old woman, were consistent with *myotonia congenita*, she was also referred for DNA testing for myotonic dystrophy. She suffered from muscle stiffness, myalgia, myotonic discharges, light myogenic changes of action potentials of motoric units and bilateral incipient subcapsular cataract. Analyses revealed no DM1 expansion, however, an uninterrupted 37 CCTG-containing *CNBP* allele with a complex repeat tract length of 208 bp was identified in her genome. Since further analyses revealed a homozygous *CLCN1* nonsense variant (NM_000083.2:c.2680C > T; p.Arg894*) in the patient’s genome and the same 37 CCTG-containing *CNBP* allele in her 68-year-old mother, who was reported to be asymptomatic, pathogenic potential of the particular *CNBP* allele seems to be unlikely. Still, this allele is interesting due to its inheritance pattern. It was transmitted to the index patient from her mother without repeat number changes, while one-unit contraction to 36 CCTG repeats occurred during the transmission to a presently 22-year-old son of the index patient. His only manifestation at the time of blood sampling was unilateral ptosis while both this son and the mother of the proband were found to be heterozygous carriers of the p.Arg894* *CLCN1* variant.

An allele with 42 CCTG repeats, as part of a 226 bp complex tract, was also mentioned in our previous report [6], however, at that time we were not able to provide extended family analysis and identify any alternative causative pathogenic variant in the family. The carrier is a 40-year-old, yet asymptomatic, woman belonging to family_4, who subjectively still does not report any neuromuscular or other health problem except for being unable to conceive. Our present findings revealed that the allele was likely to have been paternally inherited, however, paternal DNA is still not available for analysis. The initial identification of the allele of interest was made in a DNA sample deposited 16 years before the present, while repeated blood sampling and DNA isolation revealed a one-unit addition to the CCTG repeat (230 bp complex tract) over this time, suggesting certain time-dependent somatic instability, at least in leukocytes. The only affected family member, the younger sister, suffers from polyneuropathy, most likely an autosomal recessive distal spinal muscular atrophy (DSMA2; OMIM#605726) with an identified likely pathogenic sigma non-opioid intracellular receptor 1 (*SIGMAR1*) variant in a homozygous state, which is in line with clinical, molecular and genealogical findings in the patient/family (Figure 1). The uninterrupted allele of interest was, however, not identified in her *CNBP* gene, thus proving that it cannot be responsible for her neuromuscular complaints. Stability of the allele during intergenerational transmission could not be assessed in this case because of a lack of paternal DNA samples as well as offspring of the carrier women.

When assessing the likely origin of the uninterrupted likely premutation-range alleles, we genotyped the rs1871922 single nucleotide variant, which was previously found to have the same allele in all DM2 expanded alleles, even in our population [6]. All the above reported uninterrupted grey-zone alleles were found to be associated with the C allele of the rs1871922 marker, or at least this association was not disproved if the phasing was not explicitly assessable (Appendix A).

When determining the stability of other microsatellite loci in the genomes of our patients carrying uninterrupted unstable CCTG alleles, 17 other microsatellite loci tested did not show any change during intergenerational transmission (Appendix A), suggesting that the phenomenon of repeat instability in neither of our cases was a general microsatellite instability in the genomes of the carriers. Similarly, when analysing the same 17 loci after repeated blood sampling, neither of the repeated DNA analyses shows changes in allele lengths (Appendix A). 

## 4. Discussion

Regarding repeat expansion disorders, at least two main clinically relevant and mutually interconnected domains are of general interest: (i) studies on repeat length stability/instability and molecular mechanisms of the expansions itself which in turn may be important to design efficient therapeutic algorithms; and (ii) genotype–phenotype correlations with detailed threshold descriptions of pathogenicity for clinical management, prognostic and counselling purposes. 

Regarding the stability and instability, it is generally presumed that the *CNBP* repeat motifs having interrupted their CCTG parts are intergenerationally stable. Although we did not find conclusive approval of this presumption in the available literature, findings based on our analyses of several parent-to-child transmissions fully confirmed it. This agrees with the proposed hypothesis that the presence of interrupting motif(s) in the CCTG repeat tract has the potential to hinder the shifting of mini-loops which can be subsequently recognised and removed more efficiently by the proteins involved in mismatch-repair mechanisms [14]. In addition, the hypothesis that uninterrupted CCTG repeat tracts can allow the shifting of the mini-loops in both the 5′ and 3′ directions, making them more easily escape from mismatch repair mechanisms [14], seems to be supported with our findings about alleles lacking interruptions and having CCTG repeats above approximately 30 units. These were found to be intergenerationally unstable in each of our families. Although one contraction was also recorded, repeat numbers were mainly increased during transmission. Moreover, time-dependent somatic instability can be added to the attributes of uninterrupted alleles, although starting most likely at higher numbers than are required for intergenerational instability. We found such time-dependent change only in our largest identified uninterrupted allele having 42 CCTG repeats at the time of the initial sampling. All these findings suggest that the alleles described by us may constitute unstable DM2 premutation alleles.

The association of the uninterrupted grey-zone alleles with the C allele of the rs1871922 marker is in line with the hypothesis that uninterrupted alleles can contribute to the premutation allele pool in the general population with the potential to gradually extend to the mutation range when passed to successive generations. It was hypothesised that all the studied expanded CCTG alleles have a common origin [8], and that the factor creating the premutation allele pool is quite rare. It could be, for example, a special recombination event that led to the loss of the generally present sequence interruptions [7]. To function as a reservoir for hundreds of generations would require, however, those alleles created to persist in the population in a stable form. It seems likely that there should also be another factor that pushes such “normally stable” uninterrupted alleles in certain individuals over the instability threshold, transforming them into premutations. Such premutations eventually become pathogenic if they are passed to further generations and increase their lengths. In line with this, we previously reported that uninterrupted CCTG alleles are not restricted only to large unstable alleles, since they are relatively common in the general population and consistently exist in the whole spectrum of healthy-range alleles, from the smallest alleles up to the largest ones, while those under approximately 30 CCTGs are stable during intergenerational transmission (Figure 2) [6]. 

An important question is, however, what is the triggering factor for certain alleles to become intergenerationally unstable. An interesting association of a mismatch repair gene (*MSH3*) polymorphism with levels of somatic instability in DM1 patients was described. This suggested that, although alleles exceeding a certain threshold in length are inherently unstable by their length, common variants in mismatch repair genes can subtly modulate the absolute degree of instability of expanded alleles [15]. Although a similar association of common variants of mismatch repair genes to instability of smaller alleles was not reported, association of rare variants to general microsatellite instability is a relatively rare but well-known pathological phenotype associated with many cancers, specifically colorectal cancer and other types of cancers collectively known as Lynch syndrome [16]. It was suggested by Guo et al. (2016) that repeat expansions can occur if there are both strand slippage and repair escape, while strand slippage occurs more probably with increasing uninterrupted repeat length and repair escape can happen because of a less effective mismatch repair mechanism. 

It is worth noting that the phenomenon of repeat instability in neither of our cases was a general microsatellite instability, since 17 other microsatellite loci tested did not show any change during intergenerational transmission (Appendix A) or repeated DNA analyses (Appendix A). This suggested that the instability of the repeats is most likely an attribute of the *CNBP* allele structures themselves, rather than the genome. It does not rule out the possibility, however, that the original passing of the stability threshold happened in one of the ancestors of our index cases. According to this, we hypothesise that the change from a stable uninterrupted CCTG allele to an unstable one would require several preconditions coexisting in one of the predecessors of our patients (or DM2 patients in general) in whom this change occurred. Such preconditions could be, for example: (i) the presence of a stable upper-range uninterrupted CCTG allele that was possibly created by a recombination event and was maintained and distributed during several generations in a stable form; (ii) a functional variant of one of the mismatch repair genes that predisposes to microsatellite instability that could even be a general microsatellite instability; (iii) increasing repeat numbers in gametes; and (iv) transmission of the particular allele, that is now unstable by its length with no detectable general repeat instability, across further generations, in one of which it is recognised as being unstable (in our cases) or pathogenic (in the case of conventional DM2-causing alleles).

Regarding clinical significance and pathogenic potential in family_1, the identified DM1 expansion matching the patient’s symptoms and a complaint-free elderly grandmother having a one-repeat smaller allele supports that the identified uninterrupted 33 CCTG allele is most likely not disease causing. In both family_2 and 3, our findings suggest that the identified uninterrupted CCTG alleles having 34 and 37 repeats are not responsible for the patient’s symptomatology, since both mothers lacked the patients’ symptoms while carrying the same or a very similarly sized *CNBP* allele. Furthermore, a well-known homozygous *myotonia congenita*-causing mutation was identified in both index patients. Although pathogenic *CLCN1* variants are well-known modifiers of the disease phenotype in both DM1 [17] and DM2 [18,19], they are generally identified in a heterozygous state among DM patients which is normally not sufficient to cause the recessive form of *myotonia congenita*. It should be mentioned, however, that it can enhance the myotonia in DM patients as even heterozygous carriers of *CLCN1* variants can present with subclinical myotonia that is most probably combined with the expansion mediated alternative splicing disturbance of *CLCN1* in DM patients [20]. This seems to not be the case, however, in our families as both mothers, carriers of a heterozygous *CLCN1* variant with the DM2 premutation allele, were reported to be without complaints. Possible subclinical presentations were not studied in them. In family_4, no complaints in the carrier, a missing variant of interest in the affected sister and an identified likely pathogenic variant in the affected sister suggest that the grey-zone 42 repeat-containing *CNBP* allele is not responsible for the disease in the family. It should be mentioned, however, that we cannot yet assess the pathogenic potential for conditions possibly appearing later in life of the carrier and we are not able to assess whether this allele is associated with the reproductive problems of its carrier.

One could raise concerns regarding the identified cataract in family_2 and reproduction problems in family_4 as both cataracts and infertility are common symptoms connected to DM. It is important, however, that both of them exist as stand-alone health conditions in the general population without being part of a complex phenotype, while both are relatively common but highly heterogeneous pathologies with complex aetiology that includes both environmental and genetic factors [21,22]. Moreover, as they are not 100% penetrant features of neither DM1 nor DM2 (even if full expansions are present), we yet lack the studies which would assess all possible modifying genes and factors of cataract and infertility which can accentuate these symptoms in certain DM patients, as is the case with the abovementioned heterozygous *CLCN1* variants. On the other hand, it is important to note that around 35% of patients tested for DM1/DM2 in our facility, but having no identified DM1 or DM2 expansions, had reported cataract among their symptoms (unpublished data of the authors). 

Since expanded *DMPK* or *CNBP* transcripts form typical discrete nuclear foci in cells of DM1 and DM2 patients [23,24], in clinically ambiguous cases, fluorescent in situ hybridisation (FISH) has been widely utilised to confirm the diagnosis through determination of the presence or absence of nuclear foci containing expanded CUG or CCUG RNA molecules [5,25]. In context of the available literature, this approach seems to have, however, limited informational content for the determination of pathogenic potential of small alleles. It was reported that even relatively low levels of (CCUG)_100_ repeats are able to form foci in inducible cell lines, while (CCUG)_36_ repeat-containing transcripts required expression at high levels to form detectable foci [26]. In DM1 prenatal diagnostics, for example, alleles with <100 CUG repeats were reported to not form detectable foci in cultured trophoblast cells [25]. Moreover, although the foci themselves may reflect the disease and are considered for disease markers, they are likely neither causative nor essential to the disease process, since RNA foci formation, muscleblind depletion and elevated CUG-BP1 levels were found to be separable events from alternative splicing disruption [27,28,29]. All these published results led to the decision to not test the patients for the possible presence of these loci to further confirm the benign nature of these alleles.

Based on the abovementioned findings, we report basically three different situations in four families. In the first one, a causative DM1 expansion is a well-known pathogenic factor in the particular family that fully co-segregates with the disease. Besides this, in one of the affected family members, an ambiguous variant, represented by the CCTG premutation, was identified which has uncertain clinical significance, at least according to the available literature. In the meantime, this same allele was also identified in another family member but lacks co-segregation with the affection status in the family. The next two families share very similar features to each other as in both (i) the clinical signs of the probands are consistent with the diagnosis of *myotonia congenita*; (ii) there was the same *CLCN1* pathogenic variant identified in a homozygous state in the genome of the probands together with a premutation-range CCTG allele of unknown clinical significance but with a very similar size; (iii) both of them inherited one of their *CLCN1* variants and the CCTG premutation from their mother who was not reported to have signs of the disease; (iv) both of them passed one of their *CLCN1* variants and the CCTG premutation to their sons; and (v) in both families, the identified inheritance pattern of the *CLCN1* variants was in line with a recessive inheritance of *myotonia congenita*, while a co-segragation of the disease and the premutation-range CCTG alleles was not proved. The third situation is completely different from those previously mentioned and is connected to the identification of a grey-zone CCTG allele of unknown clinical significance in a healthy woman. She was tested as an unaffected family member because of her affected sister in whom, on the other hand, the CCTG allele of interest was not identified. Finally, for the classification of the identified variants regarding their most likely clinical significance, we applied the standards and guidelines of the American College of Medical Genetics and Genomics (ACMG) for interpretation of sequence variants [30]. According to these, the alleles of interest in our cases fulfil one piece of strong (“BS4: Lack of segregation in affected members of a family”) and supporting (“BP5: Variant found in a case with an alternate molecular basis for disease”) evidence of their benign nature, allowing us to classify them as “likely benign” variants, at least for the condition for which the families were molecularly examined. It should be noted, however, that although these alleles are likely to be non-pathogenic themselves, their possible modifying role on the symptoms of each affected patient cannot be explicitly ruled out. Therefore, if they are identified in a patient having stronger clinical symptomatology, it may be worth carrying out further clinical, genealogical and molecular investigations to reveal alternative genetic determinants of the phenotype of concern. 

## 5. Conclusions

In conclusion, our results further justify our previous suggestion that regarding the CCTG tract, that *CNBP* alleles can be divided into interrupted and uninterrupted alleles. Interrupted alleles are stable, while the uninterrupted group can be further divided into stable non-pathogenic alleles, unstable premutation-range alleles lacking direct clinical significance for their carriers, and unstable mutation-range alleles with pathogenic potential (Figure 2) [6]. Although it is hard to establish the precise borders for these groups based on repeat lengths alone, instability most likely begins at approximately 30 CCTG repeats and gradually rises with the increasing length of uninterrupted tracts above this threshold. Our empirical data on the approximate threshold and dynamics of instability fit well with an estimated instability threshold of 100–200 bp in repeat expansion disorders [31] as well as with the phenomenon described on cell cultures that instabilities are greater for longer tetranucleotide repeat tracts [32]. Determination of a definite generally plausible pathogenicity threshold is, however, complicated by somatic instability [5], by the modifying potential of genomic background and also by the to date undetermined nature of the entire range of DM2 symptoms [9]. Since DM2 grey-zone alleles are relatively common in the general population, careful genealogical, clinical and genetic characterisation of individual patients is extremely important to correctly assess the pathogenic potential of these alleles and to provide reliable genetic counselling.

## Figures and Tables

**Figure 1 jcm-10-03934-f001:**
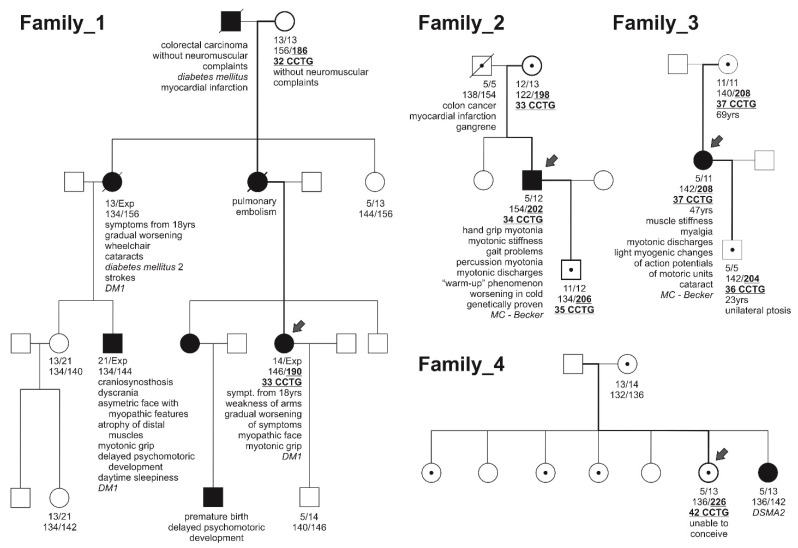
Genetic pedigree charts of the four described families. The first text line under each individual indicates the DM1 genotype and is shown in CTG repeat numbers, the second line indicates the DM2 genotype in base pairs for the complex repeat tract, while the third one shows the CCTG repeat numbers of the uninterrupted *CNBP* alleles. Individuals with no displayed numbers indicate no DNA samples available for analysis. Patients marked by dots are carriers of the identified pathogenic variants while the family members highlighted in black suffer from neuromuscular complaints: myotonic dystrophy (DM1) in family_1, autosomal recessive *myotonia congenita* in family_2 and 3 (MC–Becker) and autosomal recessive distal spinal muscular atrophy (DSMA2) in family_4. Arrows indicate individuals in whom the uninterrupted CCTG allele was initially identified (index patients), the bold and underlined numbers in the DM2 genotypes indicate the length of the uninterrupted alleles of interest and the bold connector lines show the inheritance line of these alleles through the generations.

**Figure 2 jcm-10-03934-f002:**
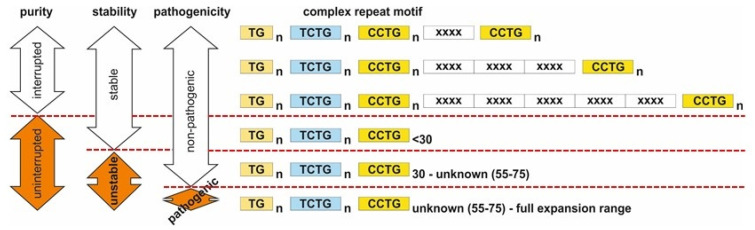
Schematic representation of the suggested division of *CNBP* alleles based on the structure of the CCTG part of the complex (TG)_n_(TCTG)_n_(CCTG)_n_ repeat tract, regarding sequence purity, inheritance and pathogenicity.

**Table 1 jcm-10-03934-t001:** Summary of findings in individuals carrying uninterrupted CCTG alleles.

Family	Individual	Sex	Age	Number of Uninterrupted *CNBP* Repeats	Length of the Complex Motif	rs1871922 Background ^#^	Nature of the *CNBP* Allele	Diagnosis (Molecularly Confirmed)
1	Index patient	F	31	33 CCTG	190 bp	A/C	Likely non-pathogenic, but unstable	DM1
	Grandmother	F	80	32 CCTG	186 bp	A/C		Without complaints
2	Index patient	M	21/26/43 *	34/34/34 CCTG	202 bp	C/C	Likely non-pathogenic, but unstable	Myotonia congenita
	Mother	F	42	33 CCTG	198 bp	A/C		Without complaints
	Son	M	2	35 CCTG	206 bp	A/C		Without complaints
3	Index patient	F	47	37 CCTG	208 bp	A/C	Likely non-pathogenic, but unstable	Myotonia congenita
	Mother	F	69	37 CCTG	208 bp	A/C		Without complaints
	Son	M	23	36 CCTG	204 bp	A/C		Without complaints (except unilateral ptosis)
4	Sister of index patient	F	24/40 *	42/43 CCTG	226 bp	C/C	Likely non-pathogenic, but unstable	Without complaints (except inability to conceive)

Family relationships are given regarding the index patient in each family. Age is given in years. Columns reporting the CCTG repeats, as well as the length of the complex motif, show only the uninterrupted allele in each individual (for information on the other identified allele, see Figure 1). In the rs1871922 background, the genotype of each patient is given, while if the phasing with the uninterrupted allele is deductible, the phased allele is marked by bold and is underlined. **^#^** Bold and underlined alleles are in *cis* phase with the uninterrupted CCTG allele. * Repeated blood sampling at different ages. F = female; M = male.

## Data Availability

Data is contained within the article or Appendix A.

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
