# Peer review of "Characterisation of Non-Pathogenic Premutation-Range Myotonic Dystrophy Type 2 Alleles"

_jcm, 2021, doi:10.3390/jcm10173934_

Round 1

Reviewer 1 Report

In this manuscript, the authors classify the CCTG repeat CNBP alleles that cause myotonic dystrophy type 2 into two main groups: interrupted and uninterrupted alleles. And suggest the stability of the repeat expansion depends on the length. This paper is very well written and organized. The materials and method section is written comprehensively, and the discussion is complete and well structure. Furthermore, the classification in the schematic representation in figure 2 helps to understand the complexity of CCTG structures in CBNP loci, which is relevant for both clinicians and basic researchers in myotonic dystrophies.

Author Response

We would like to thank the reviewer for the evaluation of our manuscript and for the positive comments.

Reviewer 2 Report

The current version is a comprehensive analysis of four targeted families. 

Include a reference to figures 1 and 2 in the text of the results - currently these figures dont appear to be linked to a specific section in the text?

The results and discussion are quite long / wordy - could you present a more complete summary of your findings in a table that includes the phenotype, inheritance, repeat summary and presence of the rs1871922 marker for each family as a summary that you can then refer to in the results and the discussion. This could reduce some of the repetition in both the results and the discussion.

The discussion can be shortened - paragraph 3 and 7 can be reduced to highlight the most important findings as these are very long sections that could benefit from additional editing to highlight the key points more clearly to the reader.

Author Response

We would like to thank the reviewer for the evaluation of our manuscript and for the valuable comments and suggestions. Based on these recommendations

  • We performed a language revision of the manuscript;
  • We performed revision of the description of the methods section to make it more clear (specifically with an emphasis on more easier reading);
  • We performed a revision of the description of the results section to make it more clear (specifically with an emphasis on easier reading). This section was supplemented by a Table according to the suggestions of the reviewer;
  • We reviewed the references for the used Figures and added them to the results section.

Reviewer 3 Report

The authors reported four families with members reporting neuromuscular complaints, in whom they identified 9 ambiguous CNBP alleles containing uninterrupted CCTG repeats in the range between 32 and 42 repeats. They revealed that these grey-zone alleles are most likely not pathogenic themselves, because of identification of other pathogenic mutations and/or family structures. Moreover, they found that these grey-zone alleles were unstable during intergenerational transmission whereas there was no observable general microsatellite instability in the genome of the carriers of these alleles. This is an interesting and important study, but I only had a few minor comments:

  • To decide the pathogenicity of ambiguous CNBP alleles, the clinical information regarding neurological symptoms and examination would be very helpful. In general, muscle weakness and waste of DM1 and DM2 are prominent in distal and proximal limbs, respectively, whereas myotonia congenita lacks muscle atrophy in the early disease stage.In patients and methods, a more precise clinical description would be useful for understanding of a pathogenic role of CNBP
  • The authors mentioned that pathogenic CLCN1 variants are well known modifiers of the disease phenotype in both DM1 and DM2. I wonder if the affected patients in family_2 and _3 were diagnosed as DM or myotonia congenita. As I mentioned above, a more precise clinical description of the patients and family members would be helpful for the decision.
  • If extramuscular symptoms, such as cardiac conduction abnormality, posterior subcapsular cataract, diabetes mellitus, and testicular dysfunction were examined in the family members, the authors should include the information.

Author Response

We would like to thank the reviewer for the evaluation of our manuscript and for the valuable comments and suggestions. Based on these recommendations

  • We performed a revision of the description of the materials and methods, results and the discussion to make the manuscript more clear and understandable. Results were also supplemented with a Table according to the suggestions of the second reviewer. The added table also contains information about the final diagnosis in the index cases. If extramuscular 
  • Information about the evidence of extramuscular symptomatology was added to the manuscript.